

# Evolutionary Characteristics of Lightning and Radar Echo Structure in Thunderstorms Based on the TRMM satellite

Xueke Wu[1], Tie Yuan[1], Rubin Jiang[2], Jinliang Li[1]

[1]College of Atmospheric Sciences and Key Laboratory for Semi-Arid Climate Change of the

Ministry of Education, Lanzhou University, Lanzhou 730000, China

[2]Key Laboratory of Middle Atmosphere and Global Environment Observation (LAGEO), Institute

of Atmospheric Physics, Chinese Academy of Sciences, Beijing 100029, China

**Correspondence**:

Dr. Xueke Wu

College of Atmospheric Sciences, Lanzhou University

No. 222, Tianshui Southern Road, Lanzhou 730000 (P.R. China)

Email: wuxk@lzu.edu.cn



**Abstract**: Based on the 16-years Tropical Rainfall Measuring Mission (TRMM)
satellite observational data, the convective characteristics of thunderstorms over
different topographic regions, as well as their radar echo structural and lightning
activity, are analyzed. The results reveal that thunderstorms over the Tibetan Plateau
have weak lightning frequency and small horizontal scale, but their occurrence
frequency is the largest, accounting for ~20% of total precipitation events, followed
by 10% over the adjacent foothills to the east and hilly land in southern China, with
the lowest occurrence frequency (~3%) over the ocean. The 30 dBZ echo top height is
a good indicator to predict the occurrence probability of lightning in convective storm,
which is more concise and intuitive than the 20 and 40 dBZ echo top heights.
Integrating the ratio of convective rainfall to total rainfall and the three-dimensional
radar echo structure features to identify thunderstorm life-cycle stages has been
proved to be a useful method, and which can help us further explore and maximize the
usage of the valuable convective event data from non-geostationary satellites. It is
found that the development of dynamic process, which refers to radar echo vertical
structure, precedes the lightning activity during the evolution of thunderstorms.
Although both lightning activity and radar echo structure peaked at the mature stage,
thunderstorms before reaching the mature stage are stronger in radar echo vertical
structure while weaker in lightning activity, and vice versa after the mature stage.
Even during the dissipating stage of thunderstorm, there still some lightning was
observed.



## Introduction

Thunderstorms are responsible for the development and formation of many severe weather phenomena, e.g., damaging wind gusts, large hail, flash floods, lightning, and tornadoes, and usually result in serious loss of human property and lives. Moreover, thunderstorms play a vital role in near-surface water and pollutants entering the stratosphere due to vertical convective transport (Park et al., 2007; Randel et al., 2010; Qie et al., 2014). This is a hot scientific topic in recent years with global climate change. However, as thunderstorms usually occur randomly in time and space, which limits the effective detection of thunderstorms, understanding of the thunderstorm formation mechanism and forecasting of thunderstorms still requires more in-depth study.

The Tropical Rainfall Measuring Mission (TRMM) satellite (Kummerow et al., 1998, 2000) was launched in 1997 and officially ended on 15 April 2015 after the spacecraft depleted its fuel reserves. It provided groundbreaking three-dimensional (3D) images of rain and storms for 17 years, far beyond the expected 3–5 year lifetime. Using the long-term and high-quality data from the TRMM satellite, global and regional rainfall, convective systems, and thunderstorms have been widely studied, and many useful and valuable scientific results have been obtained (e.g., Nesbitt et al., 2000; Cecil et al., 2005; Zipser et al., 2006; Liu et al., 2007; Houze et al., 2015). The geographical distribution of the most intense storms (Zipser et al., 2006), deep convection (Liu and Zipser, 2005; Liu et al., 2007), and lightning (Cecil et al., 2014; Albrecht et al., 2016) over the global tropics have been investigated in



detail. These studies indicate that the deepest convection mainly occurs over the
tropics (Liu and Zipser, 2005; Liu et al., 2007), the highest lightning frequency and
most intense storms are mainly found over continental regions, and the highest
density of thunderstorms is located in the subtropics (Zipser et al., 2006; Cecil et al.,
2014; Albrecht et al., 2016). In addition, the regional, seasonal, and diurnal variations
of different types of extreme convection over subtropical South America (Rasmussen
et al., 2014), the South Asian region (Romatschke et al., 2010; Qie et al., 2014), and
the Himalayan region (Houze et al., 2007; Wu et al., 2016) have been studied. It has
been found that the most intense convection occurs upstream of and over the lower
elevations of mountain barriers (Houze et al., 2007; Wu et al., 2016), while mesoscale
convective systems with the most robust stratiform regions occur primarily in the
rainiest season and regions (Romatschke et al., 2010). All these results reveal that the
distribution of convection has obvious regional differences, which means that the
occurrence and development of convective systems are closely related to regional
atmospheric conditions and topographical features.
Many studies (Reynolds et al., 1957; Takahashi, 1978; Jayaratne et al., 1983;
Saunderset al., 1991; Bürgesser et al., 2006) have suggested that the juxtaposition of
updraft and mixed-phase microphysics (0 to −40℃) provides favorable conditions
where non-inductive charging can efficiently occur via collision and separation
between graupel/hail and ice crystals in the presence of supercooled liquid water in
thunderclouds. Therefore, lightning activity is closely linked to the dynamic and
microphysical processes of thunderclouds and it considered to be an excellent



indicator for studying convective intensity of vigorous thunderstorm (Ingersoll et al.,
2000; Deierling and Petersen, 2008; Qie et al., 2015). A rapid increase in lightning
frequency means that the cloud updraft has entered its most vigorous phase and the
intense updrafts in thunderclouds usually produce large hail, lightning, heavy rain,
tornadoes, and so on. In recent years, lightning data has become an important
supplement in the study of severe convective event with the continuous improvement
of detection technology, as well as the accumulation of high-quality lightning data.
The relationships between lightning flash rate and thundercloud radar reflectivity
structure characteristics, such as maximum echo top height (Ushio et al., 2001) and
maximum radar reflectivity at different altitudes (Cecil et al., 2005; Pessi and
Businger, 2009), have been studied. In addition, the relationship between lightning
and precipitation (Petersen and Rutledge, 1998; Takayabu, 2006; Iordanidou et al.,
2016; Zheng et al., 2016), ice-water content retrieved from radar reflectivity (Petersen
et al., 2005; Deierling and Petersen, 2008), and ice scattering signatures (85 GHz and
37 GHz polarization-corrected temperature) (Toracinta et al., 2002) have also been
studied. Based on these reported relationships between lightning and convective
properties, a variety of lightning data assimilation techniques have been explored and
applied in mesoscale forecast models, which have been shown to be effective in
improving simulation results (Mansell et al., 2007; Fierro et al., 2013; Qie et al.,
2014). It can be seen that there have been many useful results about lightning and the
convective properties of thunderstorms based on observational data from the TRMM
satellite, but, a further in-depth study of the interaction and evolution of lightning



process and dynamic and microphysical processes is still required, which can deepen
the understanding of the formation process of thunderstorms and help improve the
effectiveness and reliability of lightning data assimilation techniques. It is generally
accepted that the stronger the convective intensity of a thunderstorm, the greater the
corresponding lightning flash rate. However, the relationship between lightning flash
rate and convective intensity in some thunderstorms does not follow this pattern,
especially those convective storm events that can only observe about 80 seconds at a
time by non-geostationary satellites (i.e., the TRMM and the GPM). For example,
some storms have a strong radar echo structure (maximum reflectivity exceeding 40
dBZ) but no lightning is observed by the lightning imaging sensor (LIS), in contrast,
some storms have lightning but the maximum radar reflectivity does not exceed 40
dBZ — a similar situation also occurs in the most intense storms. This has caused
some confusion and misunderstanding of the relationship between lightning and
convective intensity in thunderstorms.
Accordingly, the purpose of the present study is to investigate the occurrence of
lightning in convective systems and the variation in the characteristics of lightning
and radar echo structure with the evolution of thunderstorms using the 16-yr TRMM
data. The data and method adopted in this study are described first. Then, the
characteristics of thunderstorms over different terrain conditions, the occurrence of
lightning in convective systems, and the pattern of lightning and intense echo core in
three types of the most intense thunderstorms are discussed. Furthermore, a schematic
is concluded and established to illustrate the patterns of lightning and radar echo



structure in different evolution stages of thunderstorms. Finally, the main conclusions
are summarized.

**2 Data and methods**
Convective systems mainly distribute in the tropics while the most intense
thunderstorms more locate in subtropical regions, their occurrence and distribution are
closely related to the atmospheric circulation and terrain conditions (Zipser et al.,
2006; Houze et al., 2007; Wu et al., 2016). Based on this, the subtropical region of
East Asia is selected as the study area in this paper. Its specific scope and topographic
features are shown in Fig. 1. The study area is further divided into four adjacent
subregions from west to east according to the different terrain conditions: the Tibetan
Plateau, the eastern foothills, the hilly land in southern China, and the ocean (mainly
the coastal ocean).
Information on the convective precipitation systems in this study was extracted
from the TRMM precipitation radar (PR) and lightning imaging sensor (LIS)
observational data from 1998 to 2013; the data in August 2001 are excluded due to the
data-quality issues associated with the TRMM satellite orbit boost (Zipser et al.,
2006). The TRMM PR provides the 3D vertical structure features and the LIS
provides the lightning flash count and view time of thunderstorms (Kummerow et al.,
1998, 2000). To statistically analyze the climatology characteristics of thunderstorms
over the study region, it is necessary to identify convective systems in the TRMM PR
orbital data. The precipitation features (PFs) from the University of Utah TRMM



database (http://trmm.chpc.utah.edu/) are adopted in this study, defined by Nesbitt et
al. (2000) and Liu et al. (2008) as contiguous TRMM PR 2A25 (Iguchi et al., 2000)
near surface raining pixels with rainfall rate > 0. After grouping PR pixels, maximum
echo top height with different reflectivities, number of PR pixels, lightning flash
counts, view time, etc., inside the PFs are calculated from the collocated orbital data.
To limit noise, only PFs with at least four contiguous PR pixels (Liu et al., 2012) are
used in this study. Some erroneous cases of PFs, e.g., abnormal and discontinuous
echo in vertical profiles (Qie et al., 2014), are also excluded. In this study,
thunderstorms are defined as PFs with at least one lightning flash observed by the LIS,
non-thunderstorms are defined as PFs without any flash observed.

Many useful results (e.g., Zipser et al., 2006; Liu et al. 2007, 2012; Qie et al.,

2014; Wu et al., 2016) have been obtained from satellite data. As is well known,
thunderstorms, regardless of type, go through three stages: an initial developing stage,
a mature stage, and a dissipation stage. Convective precipitation observed by a
satellite that operates in a non-sun-synchronous orbit can be in any of these stages.
This may have a negative impact on the relationships between lightning frequency
and the convective intensity parameters. However, most studies do not take into
account the different stages of thunderstorms when analyzing the relationship between
lightning activity and convective properties. Recently, Bang and Zipser (2015)
analyzed the distribution of the ratio of convective volumetric rainfall to total
(convective plus stratiform) volumetric rainfall based on the TRMM 2A25 product.
This study followed the logic used in previous studies (e.g., Houze, 1997;



Romatschke and Houze, 2010; Zuluaga and Houze, 2015) that as a convective system
evolves, the young, vigorous convective region matures into widespread convection
coexisting with a stratiform region, and finally into mostly stratiform precipitation. A
ratio value of 1 means that the storm is 100% convective, which is commonly typified
as 'young' convection, whereas a value close to 0 means that the dominant radar
precipitation feature (RPF) is stratiform precipitation, which is typified as 'mature'
convection (Bang and Zipser, 2015). In this study, the ratio of convective volumetric
rainfall to total volumetric rainfall and the PR echo structure characteristics is adopted
to distinguish the different stages of thunderstorms, which is beneficial for analyzing
the relationship between lightning activity and convective intensity of thunderstorms
observed by the TRMM satellite.

**3 Results**
**3.1 Thunderstorms and non-thunderstorms**

The occurrence and development of convective precipitation is closely related to

the terrain condition. Over the four terrain regions (in Fig. 1), the cumulative
distribution function (CDF) for lightning flash rate of RPFs, identified from the
TRMM orbit data, is shown in Fig.2. The occurrence frequency of thunderstorms over
the Tibetan Plateau is the highest, accounting for about 20% of the total PFs, followed
by 10% over the foothills and the hilly land. Only ~3% of PFs over the coastal ocean
have lightning. This is generally consistent with the results from a previous study (Liu
et al., 2012) which found that the ratio of the RPFs with flashes over land is 11% and



over the coastal region is 2.66%. Thunderstorms over the Tibetan Plateau are the most
frequent; however, the purple line in Fig. 2 indicates that the fraction of thunderstorms
over the Tibetan Plateau more rapidly decreases with increasing lightning flash rate
compared with the other regions. This means that thunderstorms over the Tibetan
Plateau are dominated by weak thunderstorms with less lightning, while the
occurrence of intense thunderstorms is relatively scarce. For example, the fraction of
intense thunderstorms with a lightning flash rate greater than 100 fl min$^{-1}$ is only
$2\times10^{-5}$, far less than in the other regions. In addition, the lightning flash rate values at
the three black dashed lines in Fig. 2 further confirm this conclusion. Lightning
activity of thunderstorms over the hilly land is the most active among the study
subregions, followed by the foothills region. Over the coastal ocean, although the
percentage of 3% is less than the continental regions, it is still more significant than
that over open oceans (Liu et al., 2012).

The PR echo structure characteristics of non-thunderstorms and thunderstorms

over different subregions based on the TRMM PR data are calculated and shown in
Table 1. The vertical and horizontal structures reveal that thunderstorms are
significantly taller and larger than non-thunderstorms over all the subregions. From
the maximum echo top heights of different reflectivities, especially the strong echo of
40 dBZ, it can be seen that thunderstorms over the hilly land are the most intense,
followed by the foothills. The horizontal scale of thunderstorms gradually decreases
from the ocean west to the Tibetan Plateau. Although the thunderstorms over the
Tibetan Plateau are the most frequent, the vertical and horizontal structures together





with the lightning flash rate shown in Fig. 2 indicate that thunderstorms over the
Tibetan Plateau are the smallest and weakest among the four subregions, which is
consistent with the conclusions of previous studies (Luo et al., 2011; Qie et al., 2014).
The electrification and discharge processes of thunderstorms are closely related to the
mixed-phase region of thundercloud. Accordingly, the maximum PR reflectivity
between 6 and 11 km altitude (Maxdbz6-11) is used to demonstrate radar echo
intensity characteristics in the mixed-phase region and is also listed in Table 1. Note
that the Tibetan Plateau is different from the other subregions due to its very high
terrain. The results show that the maximum reflectivity in the mixed-phase region of
thunderstorms is about 40 dBZ over the different subregions, which is significantly
greater than that of non-thunderstorms.

(Figures 2 and Table 1)


**3.2 Convective properties of thunderstorms**

The TRMM satellite runs in a non-sun-synchronous orbit, which means that the

observed data are just one moment in the life cycle of precipitation events. In other
words, those observed convective events include convective storms at all different life
stages, i.e., cumulus, mature, and dissipation stages. This may lead to confusion in
terms of understanding the relationship between lightning and convective intensity in
thunderstorms. The updraft and downdraft play an important role in the evolution of
thunderstorms from generation to maturation and eventually to dissipation. In the
initial stage of thunderstorms, convective clouds are dominated by ascending motion





and produce convective precipitation. With the evolution of the convective storms, the
ascending motion weakens, the descending motion gradually strengthens, and finally
the precipitation system is mainly stratiform precipitation when it is dominated by
descending motion in the dissipation stage. Accordingly, the ratio of convective
volumetric rainfall to total volumetric rainfall of thunderstorms as introduced in
section 2 is used to distinguish the life stage of thunderstorms observed by the TRMM
satellite. The frequency distribution of the ratio of convective rainfall to total rainfall
in thunderstorms over the four different subregions is shown in Fig. 3. The ratio over
the Tibetan Plateau is significantly different to the other regions, where the percentage
of thunderstorms with less convective precipitation is obviously higher than that over
the other subregions, and even 7% of thunderstorms do not have convective rainfall at
all. This may be due to the misidentification of rain type by the TRMM PR over the
plateau (Fu and Liu, 2007), as the TRMM PR algorithm misidentifies weak
convective rainfall events as stratiform rainfall events. Therefore, the analysis of the
ratio of convective rainfall in this study is mainly based on the other subregions, and
the ratio over the Tibetan Plateau will not be discussed hereafter, despite the fact that
the values are listed in the following tables. The peaks and median ratio indicate that
thunderstorms over the hilly land have the largest ratio of convective rainfall,
followed by the foothills. Thunderstorms over the ocean have more stratiform
precipitation compared with the continental regions. More than half of the continental
thunderstorms contain more than 80% (refer to 0.8 in the Fig. 3) convective
precipitation; this percentage over the ocean is about 70%. The variation in the



convective rainfall ratio shows that thunderstorms are mainly dominated by
convective precipitation, while only a small number of thunderstorms have less
convective precipitation, which are considered to be the thunderstorms at later stage
or even dissipation stage. This will be further discussed in detail later.
Furthermore, based on the convective rainfall ratio at different grades, the PR
vertical and horizontal characteristics of both thunderstorms and non-thunderstorms
over different subregions (Table 2) are investigated. All the results consistently
indicate that thunderstorms are significantly taller in height and larger in horizontal
scale compared with those precipitation events without lightning, regardless of the
ratio, echo intensity, or subregion. Most thunderstorms have a convective rainfall ratio
greater than 0.75, and the population of thunderstorms decreases significantly as the
ratio values decrease. This trend is particularly evident in the hilly land and foothills.
There is still a small number of thunderstorms dominated by stratiform rainfall, with
the ratio less than 0.25, which will be discussed in the following section. With the
decrease in convective rainfall ratio, the maximum 20, 30, and 40 dBZ echo top
heights of both thunderstorms and non-thunderstorms decrease; the weak echo
horizontal scales are consistently increasing while the strong echo horizontal scale
shows a different feature, which increases first and then decreases. Among the four
subregions, the different echo horizontal scales of both thunderstorms and
non-thunderstorms all consistently increase from west to east irrespective of the
convective rainfall ratio. Following the view that the ratio of convective precipitation
decreases gradually with the development and evolution of convective system, the



statistical results reveal that thunderstorms in the earlier stage are taller in vertical
profile and smaller in horizontal scale compared with the later stages of thunderstorms.
It should be noted that thunderstorms with convective rainfall ratio greater than or
equal to 0.75 are tallest over the hilly land, followed by the foothills, and then the
ocean. However, thunderstorms with a ratio less than 0.75 show a different pattern:
the echo top height decreases from the ocean to the hilly land and then to the foothills.
For non-thunderstorms, their echo top heights decrease from west to east always.

As can be seen from the results of Table 2 and Fig. 3, although most of the

thunderstorms are dominated by convective precipitation, there is still a small amount
of thunderstorms that coexist with wide stratiform rainfall, or are even dominated by
stratiform precipitation. Therefore, such weak thunderstorms (precipitation events
with lightning but the maximum PR reflectivity is less than 40 dBZ) and strong
convective events (precipitation events with maximum PR reflectivity reaches or
exceeds 40 dBZ while regardless of lightning) are further compared. The statistical
results in table 3 indicate that although there are indeed some weak thunderstorms,
their occurrence frequency over the three lower subregions is actually much lower
than that of strong convective events. In contrast, weak thunderstorms over the
Tibetan Plateau occur frequently, and the number is comparable to that of strong
convective events. This further shows the particularity of the plateau thunderstorms,
which is worth more attention in future work. The 20 dBZ, 30 dBZ and 40 dBZ radar
echo pixels counts reveal that the horizontal scale of weak thunderstorms is distinctly
smaller than that of strong convective events in all subregions, even though they are



accompanied by lightning. The PR echo top height also shows a similar characteristic:
the vertical height of weak thunderstorms is lower than that of strong convective
events over land regions. In contrast, over the ocean, the echo top heights of weak
thunderstorms are taller than those of strong convective events. To clarify this
characteristic, the vertical structure of strong convective events is further investigated:
the results show that the 20 dBZ and 30 dBZ echo top heights of strong convective
events without lightning are significantly lower than those with lightning and are also
lower than those of weak thunderstorms. The percentage of strong convective events
without lightning accounts for the total number of strong convective events being the
largest over the ocean (~90%), obviously greater than the hilly land and foothills
(~72%), which result in the echo top heights of strong convective events being lower
than those of weak thunderstorms over the ocean. It should be noted that although the
echo top heights of weaker reflectivity in such strong convective events are relatively
low, their gaps between different echo top heights (MaxH20-MaxH30 and
MaxH30-MaxH40) are smaller and the convective rainfall ratio is larger, which
reveals that such strong convective events are in the earlier developing stage. The
convective rainfall ratios of weak thunderstorms over the land regions are 0.33 over
the plateau, 0.45 over the foothills, and 0.50 over hilly land, significantly less than the
ratios of 0.65, 0.73, and 0.76 for strong convective events, respectively. Finally,
integrating the above results together, it can be concluded that weak thunderstorms
observed by TRMM satellite, in terms of smaller horizontal scale, lower echo top
height, and less convective precipitation, should be mainly thunderstorms in a later





stage or even the dissipation stage instead of isolated weak thunderstorms. This is
because an isolated weak thunderstorm is not strong enough to produce lightning in
such a weak convective intensity, with weak convective echo core (maximum
reflectivity less than 40 dBZ) and small horizontal scale.

(Figure 3 and Tables 2 and 3)


**3.3 Occurrence probability of lightning in strong convective events**

The electrification and discharge processes in thunderstorm are closely related to

the development and interaction of dynamic and microphysical processes. With the
evolution and enhancement of convective cloud, the interaction of hydrometeors
(such as ice crystal, hail, snow, and graupel) increases, and lightning discharge is
produced when the electric field in thunderclouds break through a certain threshold.
The stronger the convective intensity of a thunderstorm, the more the lightning. But,
when a convective system produces lightning is still a scientific issue. Therefore, this
section further investigates the occurrence probability of lightning in strong
convective events as discussed in the previous section (refer to table 3).

The convective intensity can be defined by the properties of the convective

updrafts in a storm (Zipser et al., 2006), but it is difficult to measure them, especially
over large areas and for long periods. According to the characteristics of convection,
more vigorous convective updraft means stronger convective intensity, which brings
more and larger precipitation particles to higher altitudes, leading to a higher echo top
height. Therefore, the echo top height is adopted as an alternative to convective



intensity. The occurrence probability of lightning in strong convective events as a
function of the maximum 20, 30, and 40 dBZ echo top heights over different
subregions are shown in Figures 4 and 5. It can be seen that the occurrence probability
of lightning in strong convective events over the foothills and hilly land is
significantly larger than that over the plateau and ocean. Owing to the higher
elevation of the Tibetan Plateau, even though the echo top height of convection is
similar to that of the other subregions, the probability of lightning is significantly
lower than in the other subregions, even lower than that for the ocean. Of course, this
phenomenon is also partly caused by thunderstorm itself being weaker over the
Tibetan Plateau with such an echo top height due to its higher elevation. The
convection distribution according to the maximum 20, 30, and 40 dBZ echo top
heights indicates that convection over the ocean is mainly characterized by lower echo
top heights, and the number of strong convective events with higher echo top heights
(e.g., maximum 30 dBZ echo height exceeding 10 km) is significantly less compared
with the other regions.

Comparing the relationship between lightning probability and maximum echo

top heights of different radar reflectivity in Figures 4 and 5, it can be seen that the
maximum 30 dBZ echo top height shows a simpler and more intuitive characteristic
compared with the 20 and 40 dBZ echo top heights. This is particularly significant
over the foothills and hilly land. Strong convective events with a 30 dBZ echo top
height less than 5 km altitude do not have any flashes basically. The occurrence
probability of lightning in strong convective events with a 30 dBZ echo top height



between 5 km and 7 km is small, less than 40%. The probability value increases with
increasing 30 dBZ echo top height. It is between 40% and 70% when the maximum
30 dBZ echo height of convection is in the range of 7–9 km, and when the height
exceeds 9 km, the probability exceeds 80%. The relationship over the ocean shows a
similar pattern with that over the foothills and hilly land; the main difference is the
smaller probability values relative to the same reference height. In contrast, the
relationship between lightning probability and the maximum echo top heights of 20
and 40 dBZ are more complex and confusing. For example, the occurrence probability
of lightning in strong convective events with 20 dBZ (40 dBZ) echo top height at 12
km (5 km) altitude shows a very wide range, covering almost all probabilities from
zero to 100%. Over the Tibetan Plateau, the occurrence probability of lightning in
strong convective events is the lowest, with almost no probabilities more than 90%,
and the relationship with the 30 dBZ echo top height is also weaker compared with
the other subregions.

Although the lightning activity has a good correlation with the convective

intensity of the convective storm, there are still many issues that need further
clarification. The probability of lightning in a strong convective event with a
maximum 30 dBZ echo top height exceeding 9 km altitude is not 100%, which means
that there are some strong convective events do not have lightning observed by the
LIS. Why? This study further calculates some statistical parameters (the count,
maximum pixels of 30 dBZ echo and ratio of convective rainfall to total rainfall) for
strong convective events with and without lightning based on the different maximum



30 dBZ echo top heights and the results are shown in Table 4. From the values listed
in Table 4, it can be seen that with increasing 30 dBZ echo top height, the count (or
percentage) of strong convective events with lightning over the four subregions
consistently increase. However, the variation in the count of strong convective events
without lightning over the different subregions shows a different pattern. The count
increases over the Tibetan Plateau while it decreases over the ocean. In the two
low-altitude land regions, the count of strong convective events without lightning is
largest when the 30 dBZ echo top height is between 5 and 7 km. The horizontal scale
(30 dBZ) of strong convective events with lightning is significantly larger than that of
strong convective events without lightning, regardless of subregion or 30 dBZ echo
top heights. A comparison of the horizontal scale and ratio of convective rainfall to
total rainfall of strong convective events with and without lightning shows that strong
convective events without lightning are significantly smaller in horizontal scale and
slightly larger in the ratio of convective rainfall compared with those with lightning.
The result clearly indicates that, in the case of similar radar echo top heights, strong
convective events without lightning may be in the pre-lightning stage or the earlier
developing stage of thunderstorms compare to those with lightning. They will
probably produce lightning if their convective intensities further enhance. It should be
noted that although it should be rare, there still be some cases where lightning was not
seen by the LIS but in fact occurred in practise.
(Figures 4 and 5, and Table 4.)



**3.4 The most intense thunderstorms**


The stronger the convective intensity of a thunderstorm, the higher the height
attained by the strong echo top (40 dBZ) and the larger the lightning flash rate. As a
result, more serious loss and damage will be caused, and the vertical upward transport
of water vapor and pollutants into the upper troposphere/lower stratosphere will be
more considerable. Therefore, in order to improve the understanding of the most
intense thunderstorms, the characteristics of lightning and dynamic processes with
evolution of the most intense thunderstorms over foothills and hilly land are further
investigated in this section. The most intense thunderstorms here refer to the top 0.1%
of convective parameters in Zipser et al. (2006), that is maximum 40 dBZ echo height
exceeding 10.5 km or lightning flash rate greater than 32 fl min$^{-1}$. Thunderstorms are
divided into three types according to the two thresholds: storm-A-type thunderstorms
are defined as those with a maximum 40 dBZ echo height exceeding 10.5 km while
with a lightning flash rate less than 32 fl min$^{-1}$; storm-B-type thunderstorms are
defined as those with both thresholds attained; and storm-C-type thunderstorms are
defined as those with a maximum 40 dBZ echo height lower than 10.5 km but with a
lightning flash rate greater than 32 fl min$^{-1}$. Statistical parameter values for the three
types of thunderstorms are listed in Table 5. The lightning flash rate together with the
maximum 20, 30, and 40 dBZ echo top heights indicate that the convective intensity
of storm-B-type is the most intense among the three types of thunderstorms, while the
horizontal scale of thunderstorms (refer to the radar echo pixels of 20, 30, and 40 dBZ
pixels), shows that storm-C-type is the largest and storm-A-type is the smallest. The



ratio of convective rainfall to total rainfall of storm-A-type is the largest (0.9),
followed by 0.85 for storm-B-type, and storm-C-type is the smallest (0.74). In
addition, the vertical spacing between the top height of different echoes (20 versus 30
dBZ and 30 versus 40 dBZ) of storm-A-type is the smallest, followed by
storm-B-type, and finally storm-C-type. The smaller vertical gaps between the top
height of different echoes means the thundercloud top structure is more compact, and
vice versa. Considering all these features together, the results indicate that
storm-B-type is the most intense among the three types of thunderstorms, with the
tallest echo top heights and the most frequent lightning activity. The three types of
convective storm are in different life cycle stages of thunderstorms according to their
convective properties. Storm-A-type, in terms of lower echo top heights, smaller
horizontal scale, lower lightning flash rate, but more compact cloud top structure, is
considered to be the pre-mature stage, younger or in an earlier stage compared with
the mature stage of storm-B-type. Conversely, storm-C-type is considered to be the
post-mature stage, which is older or in a later stage than the mature stage, with a
larger horizontal scale, less convective rainfall and more fluffy cloud top structure.
This result further confirms that using the convective rainfall ratio together with the
radar echo structures to identify the stage of thunderstorms is an effective method to
analysis the convective events observed by non-geostationary orbit satellites.
(Table 5)
**4 Lightning and echo structure patterns of thunderstorms**
It is generally considered that the electrical process and the dynamic process are



closely related in a thundercloud: the stronger the convective intensity of a
thunderstorm, the greater the accompanying lightning flash rate. However, the
statistical results in this study show that this is not the case in different stages of
thunderstorms. But no matter what, for those intense thunderstorms, they must go
through a life cycle processes from the initial trigger to the mature stage and finally
their dissipation. Based on the comparative analysis of the lightning flash rate, radar
echo structure characteristics and the convective rainfall ratio of thunderstorms from
the LIS and the PR onboard the TRMM satellite, it can be concluded that the lightning
activity lags behind the development of radar echo structure with the evolution of
thunderstorm.

A schematic diagram illustrating the coupling patterns of the radar echo structure

feature and lightning activity in different evolution stages of the thunderstorm life
cycle is shown in figure 6. In the cumulus stage (or initial developing stage) of
thunderstorms, the convective cloud is energetic and dominated by strong updraft,
when the horizontal scale is small but its vertical structure is thriving, with a strong
radar echo core (over 40 dBZ) and dense cloud top structure. Nevertheless, the
convective cloud at this stage is not or not yet strong enough to generate lightning.
This is the main reason why some convective systems observed by the TRMM
satellite have strong radar echo but no lightning. In fact, convective systems of this
kind are usually in the rapid development and enhancement stage. They will soon
develop and evolve into a mature stage of thunderstorm, characterized by high echo
top height, strong radar echo core and active lightning discharge. This also means that



thunderstorms are in the most powerful and the most destructive stage with both the
most active electrical discharge process and the most robust dynamic process, which
not only produces damage on the ground but also transports water particles to upper
troposphere or even penetrates the tropopause and directly enter the stratosphere. Note
that downdrafts caused by the drag effect of rainfall are also increasing during this
period. Then after, as the unstable energy is consumed, the updraft is weakened while
the downdraft is enhanced and begins to become dominant. As a result, the lightning
flash rate and the ratio of convective rainfall to total rainfall begin to decrease. In the
dissipation stage of a thunderstorm, the thundercloud collapses and dissipates rapidly
without the support of the updraft, the radar echo top height decreases, and the
stronger radar echo weakens more rapidly. As shown in Fig. 6, the intense echo core
weakens significantly, its maximum reflectivity is less than 40 dBZ and the echo top
structure in this stage is significantly less well organized than in the previous stages,
with larger spacing between the different radar echo tops. From the perspective of
vertical radar echo top heights and radar echo core, it reveals that convective intensity
of thunderstorms in this stage are significantly weaker than that in the cumulus stage.
The stratiform rainfall is dominant during this period while in the cumulus stage it is
dominated by convective rainfall. Nonetheless, there is still a small amount of
lightning discharges as can be seen by the LIS in this stage. This mainly results from
charge transported from the upper to lower regions of cloud by downdrafts, which can
enhance the electric field stress in and below the cloud base and further produce
lightning discharge, although the charge generating mechanisms in cloud have ceased





without the support of updrafts in the dissipation stage (Pawar and Kamra, 2013).
Therefore, some storms have lightning where the radar echo core is especially weak
with maximum radar reflectivity less than 40 dBZ. Ultimately, the thunderstorm goes
through the dissipating stage, breaking and dissipating quickly without the support of
the updraft.

More specifically, according to different patterns of convective parameters, such

as echo top heights and lightning flash rate, the mature stage of thunderstorms can be
further finely divided into three stages: 1) the pre-mature stage; 2) the mature stage;
and 3) the post-mature stage, illustrating the evolutionary characteristics of electrical
and dynamic processes with the evolution of thunderstorms. Here, the mature stage
refers in particularly the most intense stage of thunderstorms, its most typical feature
is that the updraft reaches the highest altitude. Thunderstorms at this stage have the
largest lightning flash rate, the most intense radar echo core and the highest echo top
heights. Correspondingly, in the pre-mature stage thunderstorm, all the convective
parameters of echo top height, lightning flash rate and horizontal scale are in a rapid
development and enhancement. The horizontal scale of thunderstorms in this stage is
still small, dominated by upward motion despite the downdraft also being intensified
compared to the previous stage. Dangerous weather phenomenon, such as lightning
jump, hailfall and strong wind, are most likely to appear at this stage. However, the
results from table 5 show that the lightning flash rate in this stage is significantly less
than in the mature stage, and even in the post-mature stage. In the post-mature stage,
the updraft weaken and the downdraft continues to increase and gradually begins to





dominate. As a result, echo top height, lightning activity and convective rainfall begin
to decrease but the horizontal scale, to a certain extent, still increases. Note that,
although the lightning flash rate in this stage has decreased, it is still larger than that
in the pre-mature stage, with the similar vertical echo top heights. After this, the
thunderstorm is controlled by the downdraft and begins to enter the dissipating stage
as mentioned in the previous paragraph.
(Figure 6)

**5 Conclusions and discussion**
In this study, thunderstorms over different terrain conditions in subtropical East
Asia, from the Tibetan Plateau, east to the adjacent foothills, hilly land, and finally the
coastal ocean, has been investigated using 16-year data from the TRMM satellite.
Convective parameters of lightning activity and radar structure characteristics with the
development and evolution of thunderstorms are statistical analyzed. The major
findings are summarized as follows:
The occurrence frequency of thunderstorms over the different terrain conditions
shows significant differences. The occurrence of thunderstorms over the Tibetan
Plateau is the most frequent, accounting for about 20% of the total precipitation
events observed by the TRMM PR, followed by the ~10% over foothills and hilly
land. But, the convective intensity of thunderstorms over the hilly land is the most
intense, followed by the foothills, and weakest over the Tibetan Plateau despite the
occurrence of thunderstorms being the most frequent there. The occurrence of



thunderstorms over the ocean is the least, while their horizontal scale is larger than
that over land and their convective intensity is greater than that over the Tibetan
Plateau. Both the horizontal scale and vertical height of thunderstorms are always
significantly greater than those convective events without lightning.

It is well known that the lightning flash rate and intense radar echo top height are

closely related to the convective intensity of thunderstorms: the stronger the
convective intensity, the larger the lightning flash rate and the higher the echo top
height. Nevertheless, the present study indicates that the coupling patterns of lightning
and echo top heights of thunderstorms are different in different life cycle stages of
thunderstorms. This will cause some negative effects when considering convective
intensity or analyzing the correlation between lightning flash rate and radar echo top
heights of thunderstorms, especially for those observed by non-geostationary orbit
satellites. This study confirms that, combining the ratio of convective rainfall to total
rainfall with the three-dimension radar echo structure of convective event provides be
a valuable method to distinguish the stage of different thunderstorm fragments. Those
strong convective events with a maximum radar reflectivity exceeding 40 dBZ but no
lightning, are identified as thunderstorms in the initial developing/cumulus stage
according to characteristics of more convective rainfall, smaller horizontal scale, and
more compact cloud top structure. In contrast, those weak thunderstorms with
lightning but especially weak radar echo core (maximum reflectivity less than 40 dBZ)
in terms of less convective precipitation, lower echo top height, and larger vertical
spacing between different echo tops illustrate that they are actually thunderstorms in



the dissipating stage.

In order to explore when a convective event can produce lightning, this study

further investigated the occurrence probability of lightning in strong convective
events with different convective intensity. The results reveal that for those strong
convective events with maximum reflectivity exceeding 40 dBZ, the maximum 30
dBZ echo top height shows a more concise relationship with the occurrence
probability of lightning in strong convective storms compared with the maximum 20
and 40 dBZ echo top heights. When the maximum 30 dBZ echo top height of strong
convective events exceeds 9 km, the occurrence probability of lightning exceeds 80%.
When the maximum 30 dBZ echo top height is in the range of 7-9 km, the probability
is between 40% and 70 %. Almost no lightning occurs in strong convective events
with a 30 dBZ echo top height lower than 5 km altitude. The result is of great
significance for the probability forecast of lightning in strong convective storms,
which will be very useful for lightning nowcasting and warning services. Those
strong convective events with a similar 30 dBZ echo top height but no lightning,
which are characterized by smaller horizontal scale and more convective rainfall, are
considered to be thunderstorms in an earlier stage compared with those with lightning.

Based on statistical and comparative analysis from the 16-year TRMM satellite

data, this study further summarizes the patterns of lightning activity and dynamic
processes with the evolution of thunderstorms. The results indicate that the evolution
of lightning activity lags behind the development of vertical radar echo structure
during the entire life cycles of thunderstorms. When a thunderstorm is in the mature



stage, its convective intensity parameters such as lightning flash rate and strong radar
echo top heights all reach their peak levels. But, before a thunderstorm reaches its
mature stage, it is dominated by updraft and shows energetic convective developing
features in terms of larger convective rainfall ratio, smaller horizontal scale but
stronger radar echo core and more compact cloud top structure. Conversely, when a
thunderstorm has gone through its mature stage, it is dominated by downdrafts and
appears to be weak in radar echo structure, featuring a larger horizontal scale, less
convective rainfall ratio, and more fluffy cloud top structure. Furthermore, for some
thunderstorms in the developing stage, although the radar echo top can reach a higher
altitude, there may still be no lightning. Conversely, thunderstorms that are in the
dissipating stage, although the convective intensity has been significantly weakened,
may still have lightning.

The non-geostationary-orbit satellites (i.e., the TRMM, the GPM and so on) have

provided plenty of valuable observational data for studying the climatic characteristics
of precipitation and convective systems over larger areas and even global scale over a
long period. The present study has demonstrated that using the ratio of convective
rainfall to total rainfall together with the radar echo top structure to identify the
evolution stage of thunderstorms is a useful method for analyzing the TRMM data.
This method should be a valuable reference in analyzing time discontinuous
observation data provided by non-geostationary-orbit satellites or some ground-based
instruments. It can help us further explore and maximize the use of those valuable
observation data. In addition, in some cases, only the edge portion of convective



storms are observed by satellites as the convective core is located outside of the
scanning range, which can more or less adversely affect the results of statistical
analysis and should be paid more attention in future studies.

**Acknowledgment:** The authors gratefully acknowledge the University of Utah for
providing the TRMM database via their website (http://trmm.chpc.utah.edu/). This
research was supported jointly by the National Natural Science Foundation of China
(41605001), the National Key Basic Research and Development (973) Program of
China (2014CB441406) and the Fundamental Research Funds for the Central
Universities lzujbky-2015-12.

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



**Figures and Tables**

**Figure 1**




Figure 1 Location and geographic elevation of the four subregions in this study. The
Tibetan Plateau is the area in the dashed box, with elevation greater than 3000 m. The
foothills is the area in the solid box, with elevation lower than 3000 m. The hilly land
is the continental area in the dash dot box and the sea is the oceanic area in the dash
dot box. RPFs over islands are excluded in this study.



**Figure 2**

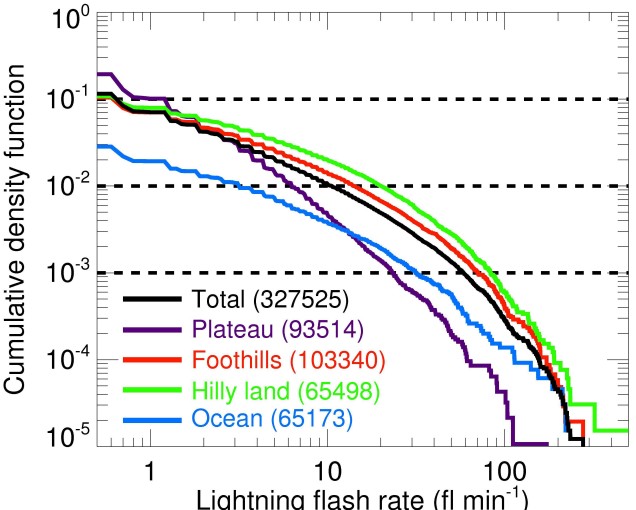


Figure 2. Cumulative distribution function (CDF) for lightning flash rate of RPFs over

the four different subregions. Sample size is given in parentheses.




**Figure 3**

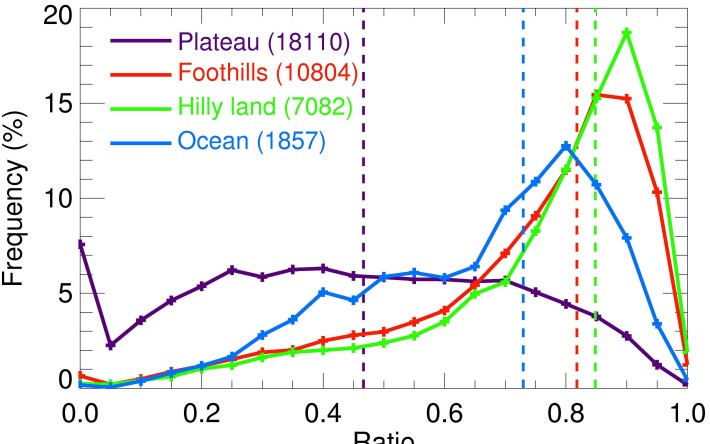

Figure 3 Frequency distribution of the ratio of convective rainfall to total rainfall of

RPFs with lightning (thunderstorms) over the four different subregions. Sample size is

given in parentheses and dashed lines represent the median ratio.



**Figure 4**

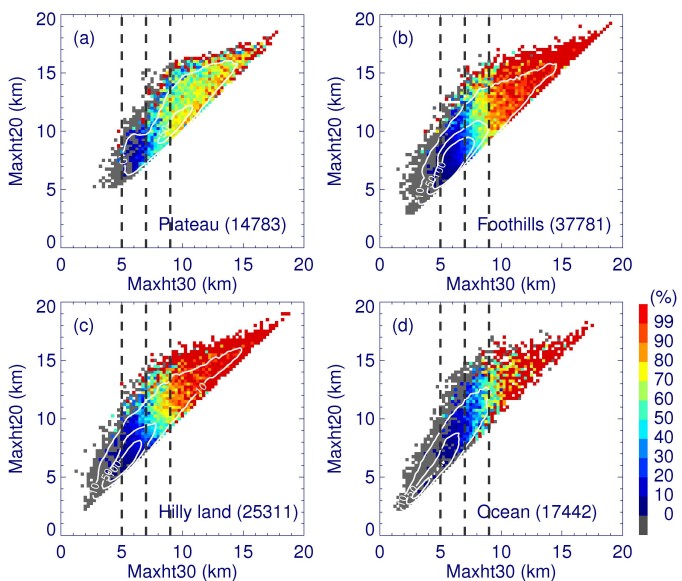

Figure 4. Occurrence probability of lightning in strong convective events with

different 20 and 30 dBZ echo top heights over the (a) plateau, (b) foothills, (c) hilly

land, and (d) ocean. White contours show the sample density of strong convective

events.





**Figure 5**

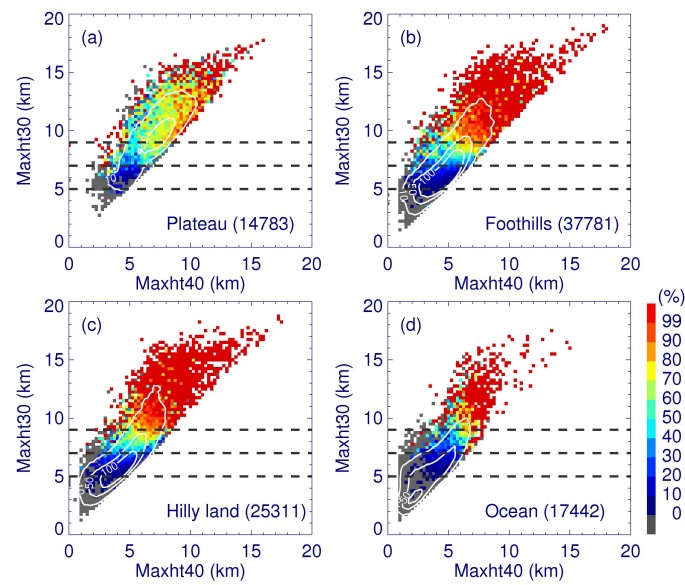

Figure 5. Same as Figure 4, but for 30 and 40 dBZ echo top heights.



**Figure 6**

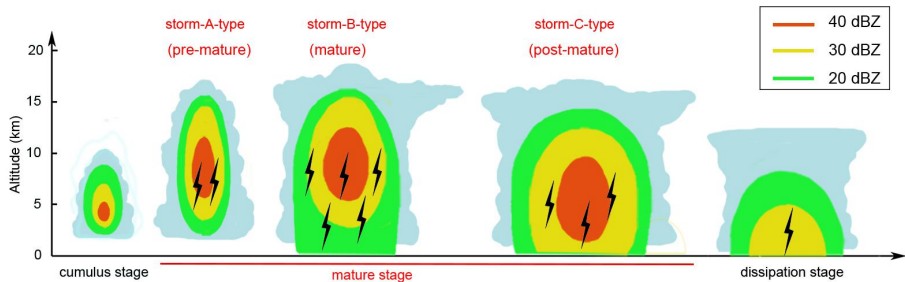

Figure 6. Schematic diagram of lightning activity and three-dimensional structural

feature in several different stages of the thunderstorm lifecycle based on the TRMM

satellite data. Light blue shades are cloud body profiles.



**Table 1**

Table 1 Statistical values of thunderstorms and non-thunderstorms over the four

different subregions.

| Subregion | Storm type | Count | Maximum height | | | Pixels number | | | Maximum Reflectivity |
|---|---|---|---|---|---|---|---|---|---|
| | | | 20 dBZ | 30 dBZ | 40 dBZ | 20 dBZ | 30 dBZ | 40 dBZ | |
| Plateau | Non-thunderstorm | 75404 | 9.2 | 7.2 | 0.6 | 21.3 | 3.3 | 0.1 | 31.7 |
| | Thunderstorm | 18110 | 11.1 | 9.3 | 3.5 | 59.8 | 12.3 | 1.4 | 38.8 |
| Foothills | Non-thunderstorm | 92536 | 6.5 | 4.9 | 1.3 | 55.7 | 14.8 | 0.8 | 19.9 |
| | Thunderstorm | 10804 | 11.3 | 9.2 | 6.1 | 202.5 | 96.2 | 17.1 | 40.5 |
| Hilly land | Non-thunderstorm | 58416 | 5.8 | 4.2 | 1.2 | 71.4 | 19.9 | 1.2 | 14.1 |
| | Thunderstorm | 7082 | 11.9 | 9.6 | 6.4 | 297.1 | 151.1 | 28.0 | 41.0 |
| Ocean | Non-thunderstorm | 63316 | 5.1 | 3.6 | 0.8 | 70.0 | 22.4 | 1.8 | 10.1 |
| | Thunderstorm | 1857 | 11.8 | 9.2 | 5.9 | 659.8 | 349.2 | 62.6 | 39.9 |


**Table 2**

Table 2. Comparison of radar echo structure characteristics between thunderstorms and non-thunderstorms at different ratios of convective rainfall to total rainfall. The vertical and horizontal characteristics are shown by maximum echo top height and the maximum pixel number of 20, 30 and 40 dBZ, respectively.

| Ratio | | | 1.00~0.75 | | 0.75~0.50 | | 0.50~0.25 | | 0.25~0.0 | |
|---|---|---|---|---|---|---|---|---|---|---|
| lightning | | | Non-thunderstorm | Thunderstorm | Non-thunderstorm | Thunderstorm | Non-thunderstorm | Thunderstorm | Non-thunderstorm | Thunderstorm |
| Plateau | | Count | 5125 | 3161 | 11760 | 5179 | 19975 | 5531 | 38544 | 4239 |
| | | Maxht20/30/40 | 10.5/8.7/3.9 | 12.1/10.4/6.7 | 9.6/7.8/1.4 | 11.7/9.9/5.2 | 9.2/7.3/0.4 | 10.8/9.0/2.5 | 8.8/6.8/0.1 | 10.0/8.1/0.4 |
| | | N20/30/40 | 11/4/1 | 31/12/3 | 13/4/0 | 61/16/2 | 21/4/0 | 75/14/1 | 26/2/0 | 60/5/0 |
| Foothills | | Count | 26346 | 6786 | 16870 | 2493 | 14865 | 1159 | 34455 | 366 |
| | | Maxht20/30/40 | 6.9/5.5/2.8 | 11.6/9.6/6.6 | 6.6/5.0/1.5 | 11.0/8.8/5.7 | 6.4/4.7/0.8 | 10.7/8.2/5.0 | 6.3/4.4/0.3 | 9.2/6.7/2.7 |
| | | N20/30/40 | 12/6/1 | 69/38/13 | 23/8/1 | 244/117/21 | 52/17/1 | 703/324/31 | 106/24/1 | 814/310/16 |
| Hilly land | | Count | 21418 | 4915 | 8947 | 1361 | 7056 | 628 | 20995 | 178 |
| | | Maxht20/30/40 | 6.0/4.7/2.2 | 12.2/10.0/6.7 | 5.9/4.4/1.4 | 11.4/9.0/5.9 | 5.7/4.0/0.7 | 10.9/8.4/5.6 | 5.7/3.8/0.3 | 9.4/6.7/4.2 |
| | | N20/30/40 | 13/6/1 | 83/48/17 | 28/11/1 | 432/226/44 | 77/27/2 | 1341/654/76 | 147/36/1 | 1507/656/49 |
| Ocean | | Count | 31127 | 856 | 11009 | 623 | 7446 | 330 | 13734 | 48 |
| | | MaxH20/30/40 | 4.6/3.4/1.0 | 11.8/9.4/5.9 | 5.8/4.1/1.1 | 12.1/9.3/6.0 | 5.4/3.7/0.8 | 11.6/8.7/5.8 | 5.4/3.6/0.3 | 10.1/7.0/4.9 |
| | | N20/30/40 | 14/6/1 | 148/85/26 | 45/18/2 | 684/371/74 | 139/56/6 | 1707/885/130 | 179/46/2 | 2283/1101/111 |





**Table 3**

Table 3 Mean convective parameters of strong convective events (with maximum

radar reflectivity exceeding 40 dBZ while regardless of lightning) and weak

thunderstorms (with lightning but maximum radar reflectivity less than 40 dBZ) over

the four different subregions.

| | | Count | Maximum height | | | Pixels count | | | Ratio |
|---|---|---|---|---|---|---|---|---|---|
| | | | 20 dBZ | 30 dBZ | 40 dBZ | 20 dBZ | 30 dBZ | 40 dBZ | |
| Plateau | Weak thunderstorm | 10146 | 10.2 | 8.5 | / | 33 | 5 | / | 0.33 |
| | Strong convection | 14783 | 11.5 | 9.6 | 7.0 | 75 | 19 | 2 | 0.65 |
| Foothills | Weak thunderstorm | 595 | 8.3 | 6.4 | / | 36 | 6 | / | 0.45 |
| | Strong convection | 37781 | 8.8 | 6.9 | 4.9 | 143 | 56 | 7 | 0.73 |
| Hilly land | Weak thunderstorm | 115 | 7.7 | 5.0 | / | 27 | 8 | / | 0.50 |
| | Strong convection | 25311 | 8.5 | 6.7 | 4.5 | 191 | 79 | 11 | 0.76 |
| Ocean | Weak thunderstorm | 32 | 8.1 | 5.7 | / | 23 | 9 | / | 0.64 |
| | Strong convection | 17442 | 7.3 | 5.5 | 3.5 | 254 | 107 | 13 | 0.74 |



**Table 4**

Table 4 Count, average of the maximum 30 dBZ echo area (Area30) and ratio of convective rainfall to total rainfall (Ratio) of precipitation events for different 30 dBZ echo top height over the four subregions.

| | | 0−5 km | | | 5−7 km | | | 7−9 km | | | 9 km ~ | | |
|---|---|---|---|---|---|---|---|---|---|---|---|---|---|
| | | count | Area30 | Ratio | count | Area30 | Ratio | count | Area30 | Ratio | count | Area30 | Ratio |
| Plateau | Non-thunderstorm | 94 | 17 | 0.57 | 1719 | 30 | 0.55 | 1830 | 13 | 0.67 | 3176 | 8 | 0.72 |
| | Thunderstorm | 3 | 53 | 0.43 | 121 | 29 | 0.57 | 1542 | 19 | 0.60 | 6298 | 23 | 0.65 |
| Foothills | Non-thunderstorm | 3963 | 38 | 0.68 | 17659 | 40 | 0.71 | 5260 | 37 | 0.78 | 690 | 22 | 0.84 |
| | Thunderstorm | 33 | 52 | 0.59 | 1266 | 79 | 0.68 | 3802 | 84 | 0.75 | 5108 | 120 | 0.81 |
| Hilly land | Non-thunderstorm | 5509 | 49 | 0.74 | 9933 | 51 | 0.74 | 2536 | 60 | 0.81 | 366 | 40 | 0.86 |
| | Thunderstorm | 36 | 54 | 0.57 | 853 | 155 | 0.68 | 2254 | 162 | 0.76 | 3824 | 149 | 0.83 |
| Ocean | Non-thunderstorm | 7167 | 37 | 0.79 | 6429 | 95 | 0.70 | 1738 | 165 | 0.71 | 283 | 196 | 0.72 |
| | Thunderstorm | 11 | 102 | 0.57 | 226 | 210 | 0.64 | 686 | 341 | 0.67 | 902 | 406 | 0.70 |



1 **Table 5**

3          Table 5. Statistical characteristics of the most intense thunderstorms over the

4                              foothills and hilly land.

| Type | Taller 40 dBZ | More lightning | Count | FlRate | Maximum height | | | Pixels number | | | Ratio |
|---|---|---|---|---|---|---|---|---|---|---|---|
| | | | | | 20 dBZ | 30 dBZ | 40 dBZ | 20 dBZ | 30 dBZ | 40 dBZ | |
| Storm-A-type | Yes | No | 359 | 14 | 15.3 | 14.4 | 11.8 | 116 | 63 | 24 | 0.90 |
| Storm-B-type | Yes | Yes | 261 | 82 | 16.1 | 15.2 | 12.3 | 542 | 327 | 107 | 0.85 |
| Storm-C-type | No | Yes | 474 | 55 | 14.1 | 12.6 | 9.6 | 1011 | 590 | 133 | 0.74 |

