# Peer review of "Evolutionary Characteristics of Lightning and Radar Echo Structure in Thunderstorms Based on the TRMM satellite"

_Atmospheric Chemistry and Physics, 2018_

## Referee Comment (RC1) · Anonymous Referee #1 · 5 Sep 2018

**Evolutionary Characteristics of Lightning and Radar Echo Structure in Thunderstorms based on the TRMM Satellite**

This manuscript aims explore the TRMM PR radar characteristics of RPFs with and without lightning over the steep terrain gradient from the Himalayan Plateau east to the South/East China Sea, and examining these characteristics from the perspective of convective lifecycle.

From a composition standpoint, there are numerous construction and grammar errors, and several instances of text copied from other articles, teetering on plagiarism. Although much of the authors' scientific argument hinges on the classification of RPFs into different lifecycle stages, the authors do not use consistent nomenclature throughout the text, which becomes exceedingly confusing.

From a scientific standpoint, I have strong reservations about using the convective/total precipitation ratio to classify storms by lifecycle stage, especially when comparing storms with lightning and without, and over diverse terrain. Convective mode will vary extremely widely by geographic regime, and the 'maturity' of storms in each regime is not universal.

Throughout the text there are many instances of gross understatement and overgeneralization of electrified convection and orographic enhancement/forcing of precipitation, and arguments made for the primitivity of the field of meteorology in these areas that is very much so not the case now. I have grave concerns about this manuscript's composition, scientific argument, and methodology, for these reasons, I find this manuscript to be substandard for this publication and suggest rejection.
* * *
**Major/Thematic Comments:**

The use of the convective/total precipitation ratio as a rough measure of an MCS's maturity dates back to Houze's 1997 paper, but its application requires the inherent assumption that you are comparing features of a same morphology, and perhaps even stricter, only comparing MCSs at different lifecycle stages. Bang and Zipser (2015) employ this ratio on RPFs with lightning only. To apply that methodology to very different geographical regimes and apply it to the entire convective spectrum is misleading. And, while I strongly do not believe that you should, there is not a point in the manuscript where you use the differing ratios explicitly to confine the RPFs into lifecycle stages, despite mentioning the different stages multiple times. This method of determining maturity is flawed, and even then, is not described clearly. I suggest looking at the recent works of Roca, Fiolleau, and Bouniol for an alternate approach to describing the lifecycle of convection.

Satellites in low earth orbit, such as TRMM, only get an instantaneous 'snapshot' of the precipitation, and therefore there are likely many cases of RPFs that were electrified and did produce lightning, especially over the ocean, where flashrates are low, that were not

observed by the LIS. This is likely not very rare, as the authors argue, and may cause problems with this line of argument.

In examining the radar reflectivity heights and profiles of RPFs with and without lightning over terrain, and over land vs. ocean, this manuscript provides little new insight beyond what Zipser, Cecil, and Liu have done in years past with the TRMM PF/RPF dataset. I find little new scientific progress accomplished by this manuscript, nor do I believe the work is a good fit for the ACP journal.

**Composition Comments:**

Throughout the manuscript, especially in the earlier introduction and methodology sections, there are large portions of text that appear from their sources (cited or otherwise) largely without or with barely minimal paraphrasing. I present a handful of examples below:

**3. 24** "Thunderstorms are responsible for the development and formation of many severe weather phenomena"

*"Thunderstorms are responsible for the development and formation of many severe weather phenomena."*
https://en.wikipedia.org/wiki/Thunderstorm

**4.54** "... mesoscale convective systems with the most robust stratiform regions occur primarily in the rainiest season and regions"

*"Convective systems exhibiting broad stratiform regions occur primarily in the rainiest season and regions"*
*(Romatschke et al., 2010; p. 419)*

**4.62** "... the juxtaposition of updraft and mixed-phase microphysics (0 to −40C) provides favorable conditions where non-inductive charging can efficiently occur via collision and separation between graupel/hail and ice crystals in the presence of supercooled liquid water in thunderclouds"

*"This juxtaposition of updraft and mixed phase microphysics provides a region where noninductive charging can efficiently occur via rebounding collisions between graupel/hail and ice crystals in the presence of supercooled liquid water"*
*(Deierling and Peterson, 2008; p. 16210)*

**8.154** "This study followed the logic used in previous studies (e.g., Houze, 1997; Romatschke and Houze, 2010; Zuluaga and Houze, 2015) that as a convective system evolves, the young, vigorous convective region matures into widespread convection coexisting with a stratiform region, and finally into mostly stratiform precipitation. A ratio value of 1 means that the storm is 100% convective, which is commonly typified as 'young' convection, whereas a value close to 0 means that the dominant radar precipitation feature (RPF) is stratiform precipitation, which is typified as 'mature' convection."

*"A value of 1 means that the RPF is 100% convective, which is commonly typified as "young" convection, whereas a value close to 0 means that the RPF is dominantly stratiform precipitation, which is typified as "mature." This logic is put forth in numerous papers [e.g., Houze, 1997; Romatschke and Houze, 2010; Zuluaga and Houze,2015] in that as a mesoscale convective system (MCS) evolves, the young, vigorous convective region matures into widespread convection coexisting with a stratiform region and finally into mostly stratiform precipitation."*

(Bang and Zipser, 2015; p 6845)

Several of these instances were detected using the similarity software, as below:

**(e.g., Houze, 1997;** 153 **Romatschke and Houze, 2010; Zuluaga and Houze, 2015) that as a convective system** 154 **evolves, the young,** vigorous **convective region matures into widespread convection** 8 **coexisting with a stratiform region, and finally into mostly stratiform precipitation.**

I do not believe that inserting the word "vigorous" into an otherwise unchanged sentence - including the references - constitutes paraphrasing.

**16.325** "The convective intensity can be defined by the properties of the convective updrafts in a storm"

*". . . intensity can be defined by the properties of the convective updrafts in a storm . . ."*
*(Zipser et al., 2006; p. 1060)*

**Other Composition Comments:**

There are massive overgeneralizations in the text, such as thunderstorms usually occur randomly in time and space," which is simply not true. Also comments like "the stronger the convective intensity of a thunderstorm, the more the lightning," made without citation or context, is a gross overstatement. ". . . lightning discharge is produced when the electric field in thunderclouds break through a certain threshold" is also a massive oversimplification of the noninductive ice-ice collision mechanism and the dieletric threshold.

Despite a large portion of scientific argument resting upon lifecycle classification, and it not being made explicitly clear as to how this classification is conducted using the convective/total ratio – the naming convention of the lifecycle stages is not consistent throughout this manuscript. The first stage is described as "initiation," "cumulus," and "triggering" and at one point, "mature" is broken up into "pre-mature, mature, and post-mature," which already is confusing, an add to the fact that the original term is included within the new term's subdivision. This makes it very difficult for readers to deduce the points you are trying to make about each.

The authors use the terms "non-sun-synchronous" and "non-geostationary" interchangeably to describe the low earth orbit of TRMM, and while TRMM is in fact neither of those types of satellite, they also mean two very different things for spaceborne observations.

I find the phrase "hilly land" to be too colloquial for formal scientific writing.

The figures, in general, are well-made and well-captioned. I would suggest in future to make the panels of figures such as 4 and 5 larger, as at present they are too small for readers' eyes to resolve the white count contours and the finer detail in the gradients of the probability of lightning colors.

---

## Referee Comment (RC2) · Anonymous Referee #2 · 19 Dec 2018

This paper presents a statistical analysis of data from 16-years of TRMM precipitation radar and lightning imaging sensor for four regions with different terrain in subtropical East Asia. The results show different characteristics of the thunderstorms in the regions. They also suggest that the 30 dBZ echo top height is a good indicator for the occurrence of lightning in the regions.

It is important to provide analysis of such long-term datasets in order to compare with models of thunderstorms. Ultimately the results will contribute to an understanding of the physical processes.

Major comments.

[Figure]
1. There is no presentation of errors in e.g. the echo intensities, the altitude of the tops of echoes or the location and timing of the two signals (lightning and reflectivity).

2. Physical explanations should be presented for the results obtained. This is important because there are some differences between the current and previous results. There is little, or no discussion of these differences.

3. The results are discussed in the final section without reference to previous work, even when a different result is obtained. The discussion section should include such references.

4. There is very little new knowledge about thunderstorms and lightning in the paper. There are in fact several papers that present similar data, but with far more extensive analysis, albeit for different geographical regions.

Specific Comments.

Attention should be paid to the English throughout the manuscript. Words are often used incorrectly.

Lines 11-14. The sentence is not clear. Useful method for what?

Lines 30-33. It is not true that thunderstorms usually occur randomly in space and time. The initiation of convection depends on several phenomena that are becoming better understood. Convection that forms over mountains for example can be quite predictable.

Lines 65-68. It is not clear why lightning is an excellent indicator for studying convective intensity. Do you mean that the lightning frequency is related to the intensity of the convection, for example?

Lines 77-83. The results of these studies should be discussed.

Lines 86-91. The sentence should be more specific by stating what results have been found and what understanding is still required. The word "formation" is not correct.

Lines 96-102. There are many physical reasons for the situations described, likely including errors in the reflectivity values. None of these are discussed and really should be.

Lines 120-123. The advection of storms from one region to the other (e.g. from land to ocean) should be discussed somewhere.

Line 135. There should be a discussion of errors and their affect on the results associated with the values of the reflectivity, altitudes, locations and timing, as well as flash rates.

Line 142. There should be as much description of the properties of the clouds in the four regions as possible. Typical soundings should be shown for the days when there is convection. Important information includes estimated cloud base temperature, altitude of the 0 and -20C levels, and typical depth of clouds for example.

Line 143-144. The reader would be interested to know more about the results. Otherwise, I suggest deleting this sentence and ones like it.

Lines 144-146. Reference should be made to Byers and Braham.

Lines 151 onward. I think it's important to discuss the main result and physical explanations of the Bang and Zipser paper. There is no discussion of an MCS for example. The current discussion does not capture the development of convective systems.

Line 189. It would be interesting to know results concerning convection over the Tibetan plateau from previous studies.

Table 1. The table should give the units for height and maximum reflectivity.

Lines 193-196. It would be helpful to have a discussion in the Introduction of the relevance of the height of the 40 dBZ echo that other studies have found – i.e. with reference to the charging zone, and the size and concentration of graupel. The results here could then be put in context.
Line 204. Is the last column of Table 1 Maxdbz6-11?

Lines 206-207. What is the relevance of the high terrain?

Line 206. It would be best to use the same terms consistently.

Lines 218 - 224. The text should be replaced by an appropriate reference.

Lines 231-234 and Figure 3. It is odd that the frequency of convective to total rainfall over the Tibetan Plateau increases to 7% for a value of zero. More information should be given about why there is a problem with identification of the rain type.

Lines 275-276. It is implied here (I think) that if there is strong convection with e.g. a large trailing stratiform region, that it is a weak thunderstorm, which is not true.

Lines 316-317. This has been stated already.

Lines 317-324. A more in-depth discussion of the charging process should be given in the Introduction and this section deleted.

Line 325-331. Most of what is written here is well known and obvious. Echo top heights have been used before for this type of analysis.

Lines 338-340. This point should be mentioned earlier – see comment above.

Line 348. "Intuitive" is perhaps not the correct word.

Lines 350-351. The result should be discussed relative to other studies.

Line 360. I think it's better to delete "and confusing".

Lines 360-363. Please refer to the appropriate figure.

Lines 399-402. It doesn't seem necessary to make this statement.

Table 5. Please explain the headings in columns 2-5.

Lines 414-416. Is this not the definition of storm-type B?

Lines 423-425. Is that not obvious?

Lines 425-427. Again, obvious.

Lines 429-432. Why could the thunderstorms not simply be mature thunderstorms that are slightly weaker than Storm B type due to say less CAPE?

Lines 432-434. Likewise for C-type thunderstorms. Delete "fluffy cloud top structure".

Lines 440-442. This has been stated previously.

Lines 451-481. This is all quite obvious and well documented; most of the text should be deleted.

Lines 490-512. There is nothing new in this discussion. Most of it should be deleted.

Section 5. There are three main points. 1. The current results should be discussed with reference to previous work. 2. Physical explanations should be given for the results using results from previous work. 3. The section is too long and should be cut by about half.

---

## Author Comment (AC1) · 1 Feb 2019

**Author's Response to Anonymous Referee #1**

**Anonymous Referee #1**

This manuscript aims explore the TRMM PR radar characteristics of RPFs with and without lightning over the steep terrain gradient from the Himalayan Plateau east to the South/East China Sea, and examining these characteristics from the perspective of convective lifecycle.

From a composition standpoint, there are numerous construction and grammar errors, and several instances of text copied from other articles, teetering on plagiarism. Although much of the authors' scientific argument hinges on the classification of RPFs into different lifecycle stages, the authors do not use consistent nomenclature throughout the text, which becomes exceedingly confusing.

From a scientific standpoint, I have strong reservations about using the convective/total precipitation ratio to classify storms by lifecycle stage, especially when comparing storms with lightning and without, and over diverse terrain. Convective mode will vary extremely widely by geographic regime, and the 'maturity' of storms in each regime is not universal. Throughout the text there are many instances of gross understatement and over generalization of electrified convection and orographic enhancement/forcing of precipitation, and arguments made for the primitivity of the field of meteorology in these areas that is very much so not the case now. I have grave concerns about this manuscript's composition, scientific argument, and methodology, for these reasons, I find this manuscript to be substandard for this publication and suggest rejection.

The authors are grateful to the reviewer for the criticism and comments on this manuscript. This is a very rare and precious opportunity for our young researchers to learn and progress. Which can help us always maintain an objective and rigorous scientific attitude in the future research work.

We have carefully considered about your comment, and have revised the

manuscript accordingly. The grammar errors and other existing problems have also been revised. For some other explanations, please refer to the responses to the comments below.

The followings are our responses to your comments and action taken in the revised manuscript.

**Major/Thematic Comments:**

1. The use of the convective/total precipitation ratio as a rough measure of an MCS's maturity dates back to Houze's 1997 paper, but its application requires the inherent assumption that you are comparing features of a same morphology, and perhaps even stricter, only comparing MCSs at different lifecycle stages. Bang and Zipser (2015) employ this ratio on RPFs with lightning only. To apply that methodology to very different geographical regimes and apply it to the entire convective spectrum is misleading. And, while I strongly do not believe that you should, there is not a point in the manuscript where you use the differing ratios explicitly to confine the RPFs into lifecycle stages, despite mentioning the different stages multiple times. This method of determining maturity is flawed, and even then, is not described clearly. I suggest looking at the recent works of Roca, Fiolleau, and Bouniol for an alternate approach to describing the lifecycle of convection.

**Response**: The authors are very sorry for the confusion that has brought you.

Yes, the use of the convective rain ratio is a rough measure, and it is because of this, the present paper use it in combination with the radar echo structures characteristics to distinguish the different stages of different thunderstorms, rather than using this ratio alone. In addition, this paper discussed four very different geographical regions, but each subregion is a relatively single terrain condition, the comparisons of different convection are performed within the same subregion.

Secondly, just as you mentioned that 'there is not a point in the manuscript where you use the differing ratios explicitly to confine the RPFs into lifecycle stages.' The purpose of this paper is not to specifically show a method to distinguish the lifecycle stages of thunderstorms, but to statistically analyze the lightning activity and radar echo structure in

thunderstorms based on the long-term TRMM satellite data. Actually, the convective rainfall ratio together with the radar echo structure (both of vertical and horizontal) and lightning flash rate are combined to compare the stages of thunderstorms. Moreover, it is not used to clearly identify which lifecycle stages of a convection belongs to, but only to help us compare a certain cluster of thunderstorms are earlier or later than the other cluster of thunderstorms. For example, as the red rectangular box marked in the following table (half of the Table 4 in the paper), there are 690 non-thunderstorms with 30 dBZ echo top exceed 9 km but no lightning was observed by the LIS. Although the 30 dBZ echo top is at the same range, the statistical values shows that these non-thunderstorms are characterized by larger convective rainfall ratio (0.84) but smaller in horizontal scale(22 pixels, far less than the 120 pixels of thunderstorms) than thunderstorms, all these convective characteristics together illustrate that the non-thunderstorms are in the earlier stage than those thunderstorms.

Table. Count, average of the maximum 30 dBZ echo pixels (Area30) and convective rainfall ratio to total rainfall (Ratio) of precipitation events for different 30 dBZ echo top height over the four subregions.

| Subregions | Types | 7－9 km | | | 9 km ~ | | |
|---|---|---|---|---|---|---|---|
| | | count | Area30 | Ratio | count | Area30 | Ratio |
| Plateau | Non-thunderstorm | 1830 | 13 | 0.67 | 3176 | 8 | 0.72 |
| | Thunderstorm | 1542 | 19 | 0.60 | 6298 | 23 | 0.65 |
| Foothills | Non-thunderstorm | 5260 | 37 | 0.78 | 690 | 22 | 0.84 |
| | Thunderstorm | 3802 | 84 | 0.75 | 5108 | 120 | 0.81 |
| Hilly land | Non-thunderstorm | 2536 | 60 | 0.81 | 366 | 40 | 0.86 |
| | Thunderstorm | 2254 | 162 | 0.76 | 3824 | 149 | 0.83 |
| Ocean | Non-thunderstorm | 1738 | 165 | 0.71 | 283 | 196 | 0.72 |
| | Thunderstorm | 686 | 341 | 0.67 | 902 | 406 | 0.70 |

The recent paper of Roca et al. (2017) you mentioned using the meteorological geostationary satellite date showed a simple parametric model, which is a very reliable and useful method to document the time evolution of the cold cloud shield of MCS over the tropics. I believe this will be of great help in the future use of satellite data to study MCSs. However, this paper mainly analyzes thunderstorms based on TRMM radar echo and lightning data, so it has not been adopted this time.

Roca R , Fiolleau T , Bouniol D . A Simple Model of the Life Cycle of Mesoscale Convective Systems Cloud Shield in the Tropics[J]. Journal of Climate, 2017:JCLI-D-16-0556.1.

2. Satellites in low earth orbit, such as TRMM, only get an instantaneous 'snapshot' of the precipitation, and therefore there are likely many cases of RPFs that were electrified and did produce lightning, especially over the ocean, where flashrates are low, that were not observed by the LIS. This is likely not very rare, as the authors argue, and may cause problems with this line of argument.

**Response:** Yes, the case you mentioned does exist. However, the official introduction about the TRMM LIS said that: 'The imager's field of view allows the sensor to observe a point on the Earth or a cloud for 80 seconds, **a sufficient time** to estimate the flashing rate, which tells researchers whether a storm is growing or decaying.' Therefore, it must be extremely weak if the case did produce lightning but not observed by the LIS within ~80 seconds. It is believed that such weak convection can not affect the results of this paper.

Again, this study focused on statistical analysis of long-term TRMM observation data, and attempts to characterize the evolution of lightning and radar echo structures in thunderstorms. It is aiming to explore the relationship between lightning and radar echoes during different cycle stage of thunderstorms from a statistical perspective.

3. In examining the radar reflectivity heights and profiles of RPFs with and without lightning over terrain, and over land vs. ocean, this manuscript provides little new insight beyond what Zipser, Cecil, and Liu have done in years past with the TRMM PF/RPF dataset. I find little new scientific progress accomplished by this manuscript, nor do I believe the work is a good fit for the ACP journal.

**Response:** There are indeed many studies about lightning and convective properties of thunderstorms, but this study is different. Several highlights of this paper are summarized as follows:

Firstly, the result shows that 30 dBZ echo top height has a concise relationship with the occurrence probability of lightning in convective storms, which will be very useful for lightning nowcasting and warning services.

Secondly, the result of this paper confirmed that combining the ratio of convective rainfall to total rainfall with the radar echo structure of convection is an effective and feasible method to distinguish the stage of different convection (snapshot of convection) observed by the TRMM. This can help us further explore and maximize the usage of the observation data from non-sun-synchronous satellites, e.g., the TRMM, the GPM.

On this basis, the coupling patterns of the radar echo structure feature and lightning activity with the evolution of the extreme thunderstorms are summarized and discussed according to the statistical analysis of 16-yr TRMM data. Furthermore, this study found that convection with stronger radar echo structure but less or no lightning, are considered as thunderstorms in developing/cumulus stage. While those weak thunderstorms, with lightning but especially weak in radar echo core (maximum reflectivity less than 40 dBZ) are actually thunderstorms in the dissipating stage. It is believed that a more stable and reliable relationship between lightning and convective properties of thunderstorms will be obtained if considering these different situations in advance. This is benefit to improving the lightning data assimilation techniques and simulation results (Mansell et al., 2007; Fierro et al., 2013; Qie et al., 2014).

**Composition Comments:**

Throughout the manuscript, especially in the earlier introduction and methodology sections, there are large portions of text that appear from their sources (cited or otherwise) largely without or with barely minimal paraphrasing. I present a handful of examples below:

3. 24 "Thunderstorms are responsible for the development and formation of many severe    weather phenomena"

"Thunderstorms are responsible for the development and formation of many severe weather phenomena."

https://en.wikipedia.org/wiki/Thunderstorm

**Response:** This sentence has been replaced in the revised manuscript.

4.54 ". . . mesoscale convective systems with the most robust stratiform regions occur primarily in the rainiest season and regions"

"Convective systems exhibiting broad stratiform regions occur primarily in the rainiest season and regions" (Romatschke et al., 2010; p. 419)

**Response:** The author believes that this is a normal literature citation.

4.62 ". . . the juxtaposition of updraft and mixed-phase microphysics (0 to −40C) provides favorable conditions where non-inductive charging can efficiently occur via collision and separation between graupel/hail and ice crystals in the presence of supercooled liquid water in thunderclouds"

"This juxtaposition of updraft and mixed phase microphysics provides a region where noninductive charging can efficiently occur via rebounding collisions between graupel/hail and ice crystals in the presence of supercooled liquid water" (Deierling and Peterson, 2008; p. 16210)

**Response:** Thanks for pointing out this dispute. Actually, I don't think this is a problem, this is a customary description about the non-inductive charging in atmospheric electricity, however, I still modified it to avoid unnecessary dispute.

8.154 "This study followed the logic used in previous studies (e.g., Houze, 1997; Romatschke and Houze, 2010; Zuluaga and Houze, 2015) that as a convective system evolves, the young, vigorous convective region matures into widespread convection coexisting with a stratiform region, and finally into mostly stratiform precipitation. A ratio value of 1 means that the storm is 100% convective, which is commonly typified as 'young'convection, whereas a value close to 0 means that the dominant radar precipitation feature (RPF) is stratiform precipitation, which is typified as 'mature' convection."

"A value of 1 means that the RPF is 100% convective, which is commonly typified as "young"convection, whereas a value close to 0 means that the RPF is dominantly stratiform precipitation, which is typified as "mature." This logic is put forth in numerous papers [e.g., Houze, 1997; Romatschke and Houze, 2010; Zuluaga and Houze,2015] in that as a mesoscale convective system (MCS) evolves, the young,

vigorous convective region matures into widespread convection coexisting with a stratiform region and finally into mostly stratiform precipitation."

(Bang and Zipser, 2015; p 6845)

Several of these instances were detected using the similarity software, as below:

I do not believe that inserting the word 'vigorous' into an otherwise unchanged sentence -including the references - constitutes paraphrasing.

**Response:** Originally, in order to describe the method clearly, it indeed quoted too long content from Bang and Zipser (2015), this might be not correct. Now, it has been modified in the revised manuscript. Thank you.

16.325 "The convective intensity can be defined by the properties of the convective updrafts in a storm"

". . . intensity can be defined by the properties of the convective updrafts in a storm . . ."

(Zipser et al., 2006; p. 1060)

**Response:** The author believes that this is a normal literature citation.

**Other Composition Comments:**

There are massive over generalizations in the text, such as thunderstorms usually occur randomly in time and space,"which is simply not true. Also comments like "the stronger the convective intensity of a thunderstorm, the more the lightning," made without citation or context, is a gross overstatement. ". . . lightning discharge is produced when the electric field in thunderclouds break through a certain threshold"is also a massive oversimplification of the noninductive ice-ice collision mechanism and the dieletric threshold. Despite a large portion of scientific argument resting upon lifecycle classification, and it not being made explicitly clear as to how this classification is conducted using the convective/total ratio–the naming convention of the lifecycle stages is not consistent throughout this manuscript. The first stage is described as 'initiation,' 'cumulus,' and 'triggering' and at one point, 'mature' is broken up into 'pre-mature, mature, and post-mature,' which already is confusing, an

add to the fact that the original term is included within the new term's subdivision. This makes it very difficult for readers to deduce the points you are trying to make about each.

**Response:** Yes, some sentences in the original manuscript were over generalizations in text, this has been corrected in the revised version. For the question about how the lifecycle classification is conducted please refer to the previous response to the major comment 1.

The authors use the terms 'non-sun-synchronous' and 'non-geostationary' interchangeably to describe the low earth orbit of TRMM, and while TRMM is in fact neither of those types of satellite, they also mean two very different things for spaceborne observations.

**Response:** Yes, this do cause some confusion, we have confirmed and unified it into 'non-sun-synchronous', this is consistent with the official statement.

I find the phrase 'hilly land' to be too colloquial for formal scientific writing.

**Response:** Sorry, we searched for it and found that the word 'hilly' was also used in some paper published in professional geographical journals. More importantly, we did not find a more appropriate phrase to describe the geographical characteristics of this area. So, this phrase 'hilly land' is still retained in the revised manuscript.

The figures, in general, are well-made and well-captioned. I would suggest in future to make the panels of figures such as 4 and 5 larger, as at present they are too small for readers' eyes to resolve the white count contours and the finer detail in the gradients of the probability of lightning colors.

**Response:** Thank you, we fully accept your suggestion and we must pay attention to it in future work.

---

## Author Comment (AC2) · 1 Feb 2019

**Author's Response to Anonymous Referee #2**

**Anonymous Referee #2**

This paper presents a statistical analysis of data from 16-years of TRMM precipitation radar and lightning imaging sensor for four regions with different terrain in subtropical East Asia. The results show different characteristics of the thunderstorms in the regions. They also suggest that the 30 dBZ echo top height is a good indicator for the occurrence of lightning in the regions. It is important to provide analysis of such long-term datasets in order to compare with models of thunderstorms. Ultimately the results will contribute to an understanding of the physical processes.

The authors are very grateful to the reviewer for their contributions to this manuscript, these constructive comments have allowed a substantial improvement of the manuscript. Our responses to comments and action taken are given below. Some other existing problems have also been revised.

**Major comments.**

1. There is no presentation of errors in e.g. the echo intensities, the altitude of the tops of echoes or the location and timing of the two signals (lightning and reflectivity).

**Response**: Yes, the errors and resolution information of data should be presented before it is used. However, due to the high-quality TRMM data has been widely recognized over the world already and there are numerous important papers have been published based on TRMM. Therefore, only the processing of the data and the methods used (limit the noise and exclude erroneous cases) are introduced while the description of data itself is relatively less than normal in the original manuscript. According to your comment, some related information about the TRMM data (the PR and LIS) has been added in the Data and method section in the revised manuscript.

2. Physical explanations should be presented for the results obtained. This is important because there are some differences between the current and previous results. There is little, or no discussion of these differences.

**Response**: Thank you, the necessary physical explanations have been supplemented in the revised manuscript based on your comment.

3. The results are discussed in the final section without reference to previous work, even when a different result is obtained. The discussion section should include such references.

**Response**: The discussion section has been re-organized based on your comment. Some too long and repeated statement have been cut as necessary.

4. There is very little new knowledge about thunderstorms and lightning in the paper. There are in fact several papers that present similar data, but with far more extensive analysis, albeit for different geographical regions.

**Response**: Sorry, please forgive the authors for not agreeing with this comment.

Firstly, the result shows that the 30 dBZ echo top height has a concise relationship with the occurrence probability of lightning in convective storms, which can be easily used in the lightning nowcasting and warning services.

Secondly, the result of this paper confirmed that combining the ratio of convective rainfall to total rainfall with the radar echo structure of convection is an effective and feasible method to distinguish the stage of different convection (instantaneous snapshot view of convection) observed by the TRMM. This can help us further explore and maximize the usage of the observation data from non-sun-synchronous satellites, e.g., the TRMM, the GPM.

On this basis, the coupling patterns of the radar echo structure feature and lightning activity with the evolution of the extreme thunderstorms are summarized and discussed according to the statistical analysis of 16-yr TRMM data. Furthermore, this study found that convection with stronger radar echo structure but less or no lightning, are considered as thunderstorms in

developing/cumulus stage. While those weak thunderstorms, with lightning but especially weak in radar echo core (maximum reflectivity less than 40 dBZ) are actually thunderstorms in the dissipating stage. It is believed that a more stable and reliable relationship between lightning and convective properties of thunderstorms will be obtained if considering these different situations in advance. This is benefit to improving the lightning data assimilation techniques and simulation results (Mansell et al., 2007; Fierro et al., 2013; Qie et al., 2014).

**Specific Comments.**

Attention should be paid to the English throughout the manuscript. Words are often used incorrectly.

**Response**: Thank you for your reminder, we have been carefully checked and revised the English edit of the manuscript.

Lines 11-14. The sentence is not clear. Useful method for what?

**Response**: It has been re-edited in the revised manuscript.

Lines 30-33. It is not true that thunderstorms usually occur randomly in space and time. The initiation of convection depends on several phenomena that are becoming better understood. Convection that forms over mountains for example can be quite predictable.

**Response**: Yes, the previous description was not accurate enough and which has been modified following your comments.

Lines 65-68. It is not clear why lightning is an excellent indicator for studying convective intensity. Do you mean that the lightning frequency is related to the intensity of the convection, for example?

**Response**: Yes, the lightning frequency is an indicator of convective intensity of thunderstorms. The description has been revised.

Lines 77-83. The results of these studies should be discussed.

**Response**: Revised.

Lines 86-91. The sentence should be more specific by stating what results have been

found and what understanding is still required. The word "formation" is not correct.

**Response**: Revised together with the previous comment and some incorrect description has been removed.

Lines 96-102. There are many physical reasons for the situations described, likely including errors in the reflectivity values. None of these are discussed and really should be.

**Response**: Yes, there are indeed many physical reasons as you mentioned, the revised manuscript has further supplemented. Actually, some efforts had been taken to avoid the adverse effects of the errors.

Lines 120-123. The advection of storms from one region to the other (e.g. from land to ocean) should be discussed somewhere.

**Response**: This paper is a statistical analysis of long-term and a large number of convective precipitation observed by the TRMM, therefore, the advection of storms has not been discussed in the manuscript.

Line 135. There should be a discussion of errors and their affect on the results associated with the values of the reflectivity, altitudes, locations and timing, as well as flash rates.

**Response**: Please refer to the response to the major comment 1.

Line 142. There should be as much description of the properties of the clouds in the four regions as possible. Typical soundings should be shown for the days when there is convection. Important information includes estimated cloud base temperature, altitude of the 0 and -20C levels, and typical depth of clouds for example.

**Response**: Yes, your comment is very important and useful in discussing convection. But, this paper aim to show the statistical results and the evolution characteristics of lightning and radar echo structure of thunderstorms, this section (2 Data and methods) mainly introduces the related data and method used in the study. However, we will consider your advise in the future discuss and study, thank you.

Line 143-144. The reader would be interested to know more about the results. Otherwise, I suggest deleting this sentence and ones like it.

**Response**: This sentence has been deleted in the revised manuscript.

Lines 144-146. Reference should be made to Byers and Braham.

**Response**: The reference has been added, thanks.

Lines 151 on ward. I think it's important to discuss the main result and physical explanations of the Bang and Zipser paper. There is no discussion of an MCS for example. The current discussion does not capture the development of convective systems.

**Response**: The main results and physical explanations of the Bang and Zipser (2015) is discussed in the Introduction section, this section is mainly the introduction of data and method rather than the result, thank you.

Line 189. It would be interesting to know results concerning convection over the Tibetan plateau from previous studies.

**Response**: The Tibetan plateau related result was discussed in the previous part of this paragraph, and some results of previous studies has been added there.

Table 1. The table should give the units for height and maximum reflectivity.

**Response**: The units of all the tables are checked and given in the revised manuscript.

Lines 193-196. It would be helpful to have a discussion in the Introduction of the relevance of the height of the 40 dBZ echo that other studies have found – i.e. with reference to the charging zone, and the size and concentration of graupel. The results here could then be put in context.

**Response**: Thank you, we have considered your suggestion during the revision.

Line 204. Is the last column of Table 1 Maxdbz6-11?

**Response**: Yes, it has been corrected in Table 1.

Lines 206-207. What is the relevance of the high terrain?

**Response**: Revised.

Line 206. It would be best to use the same terms consistently.

**Response**: OK, thanks.

Lines 218 - 224. The text should be replaced by an appropriate reference.

**Response**: The test has been re-edited by reference to the paper of Byers and Braham (1948).

Lines 231-234 and Figure 3. It is odd that the frequency of convective to total rainfall over the Tibetan Plateau increases to 7% for a value of zero. More information should be given about why there is a problem with identification of the rain type.

**Response**: The misidentification of tain type has beed supplemented in the revised manuscript. The explain of the misidentification from Fu and Liu (2007) is that the TRMM PR algorithm misidentifies weak convective rain events as stratiform rain events. The possible cause for this misidentification is believed to be that the freezing level is close to the surface over the plateau, so that the ground echo may be mistakenly identified as the melting level in the PR rain classification algorithm. This is believed to be the reason why there are about 7% of thunderstorm over the Tibetan plateau do not have convective rain according to the TRMM PR data.

Lines 275-276. It is implied here (I think) that if there is strong convection with e.g. a large trailing stratiform region, that it is a weak thunderstorm, which is not true.

**Response**: Sorry, this is a misunderstanding caused by the previous inaccuracy expression. It has been corrected in the revised manuscript.

Lines 316-317. This has been stated already.

**Response**: Removed.

Lines 317-324. A more in-depth discussion of the charging process should be given in the Introduction and this section deleted.

**Response**: Agree, the revised manuscript has reorganized the related content according to your comment. The necessary discussion and reference have been supplemented in the introduction. Thank you!

Line 325-331. Most of what is written here is well known and obvious. Echo top heights have been used before for this type of analysis.

**Response**: Yes, it is obvious, this section has been simplified, thanks for your comment.

Lines 338-340. This point should be mentioned earlier – see comment above.

**Response**: Sorry, it may be that the statement in original manuscript is not clear enough and cause the confusion. The previous part (comment above) is related to the ratio of convective rain to total rain in thunderstorm, which is about **thunderstorms**. The object discussed here is **strong convection,** precipitation events with maximum PR reflectivity exceeding 40 dBZ while regardless of lightning. Despite this, this part has been clarified accordingly.

Line 348. "Intuitive" is perhaps not the correct word.

**Response**: It has been corrected.

Lines 350-351. The result should be discussed relative to other studies.

**Response**: Here only introduces the result, the related discussion appears in in the following part together with the result of Table 4.

Line 360. I think it's better to delete "and confusing".

**Response**: Done.

Lines 360-363. Please refer to the appropriate figure.

**Response**: Done.

Lines 399-402. It doesn't seem necessary to make this statement.

**Response**: OK, this statement has been removed in the revised manuscript.

Table 5. Please explain the headings in columns 2-5.

**Response**: The table caption has been re-edited, thanks.

Lines 414-416. Is this not the definition of storm-type B?

**Response**: Not exactly the same. By definition, the lightning flash rate of storm-B and storm-C are both exceed 32 fl/min, and the 40 dBZ echo top height of storm-A and storm-B are both exceed 10.5 km. From the vertical characteristics of thunderstorms, a higher 40 dBZ echo top height does not mean the 20 and 30 dBZ echo top height are also higher, especially considering the different geographical features. Despite this, modifications have been made in the revised manuscript to avoid confusion.

Lines 423-425. Is that not obvious?

**Response**: Removed.

Lines 425-427. Again, obvious.

**Response**: Removed.

Lines 429-432. Why could the thunderstorms not simply be mature thunderstorms that are slightly weaker than Storm B type due to say less CAPE?

**Response**: Of course, your statement must be appear in nature if it is considered from the case study of thunderstorms. However, the purpose of this paper is try to use statistical methods to explain from the perspective of different stages of thunderstorm life cycle.

Lines 432-434. Likewise for C-type thunderstorms. Delete "fluffy cloud top structure".

**Response**: Done.

Lines 440-442. This has been stated previously.

Lines 451-481. This is all quite obvious and well documented; most of the text should be deleted.

Lines 490-512. There is nothing new in this discussion. Most of it should be deleted.

**Response**: According to these **3 comments,** this section (4 Lightning and echo structure patterns of thunderstorms) in the revised manuscript has been deleted and the related context of figure 6 has been briefly discussed in the end of section 3.4.

Section 5. There are three main points. 1. The current results should be discussed with reference to previous work. 2. Physical explanations should be given for the results using results from previous work. 3. The section is too long and should be cut by about half.

**Response**: Thank you very much for the suggestion, the Conclusions and discussion section has been reorganized in the revised manuscript according to your comment.